# Excess Iodine Consumption Induces Oxidative Stress and Pancreatic Damage Independently of Chemical Form in Male Wistar Rats: Participation of PPAR-γ and C/EBP-β

**DOI:** 10.3390/biology13070466

**Published:** 2024-06-25

**Authors:** Cristian Arbez-Evangelista, Omar Arroyo-Xochihua, Ilse Haide Ortega-Ibarra, Edú Ortega-Ibarra, Yeimy Mar De León-Ramírez, Estela Cuevas-Romero, Omar Arroyo-Helguera

**Affiliations:** 1Centro de Investigaciones Biomédicas, Universidad Veracruzana, Av. Luis Castelazo Ayala S/N, Col. Industrial Ánimas, Xalapa PC. 91190, Veracruz, Mexico; zs20000323@estudiantes.uv.mx (C.A.-E.); zs22000317@estudiatnes.uv.mx (O.A.-X.); 2Centro de Investigación en Alimentación y Nutrición, Universidad del Istmo, Carretera Transísmica Juchitán, la ventosa km. 14, La Ventosa PC. 70102, Oaxaca, Mexico; eoi@bizendaa.unistmo.edu.mx; 3Nutrition Faculty, Universidad de la Sierra Sur, Av. Universidad Col. Ciudad Universitaria, Carretera Transísmica Juchitán, la ventosa km. 14, La Ventosa PC. 70102, Oaxaca, Mexico; edu@biznedaa.unistmo.edu.mx; 4Laboratorio de Biomedicina y Salud Pública, Instituto de Salud Pública, Universidad Veracruzana, Av. Luís Castelazo Ayala S/N, Col. Industrial Animas, Xalapa CP. 91190, Veracruz, Mexico; 20161268@garzas.uatx.mx; 5Centro Tlaxcala de Biología de la Conducta, Universidad Autónoma de Tlaxcala, Tlaxcala PC. 90070, Tlaxcala, Mexico; ecuevas@uatx.mx

**Keywords:** C/EBP-β, GLUT-4, iodine, oxidative stress, pancreatic damage, PPAR-γ

## Abstract

**Simple Summary:**

An excessive consumption of iodine is associated with pancreatic damage. It is unknown whether iodine’s chemical forms from the diet are related because these are transported by active or passive transport in the cell and exert antioxidant or oxidant effects according to the chemical iodine form. This research analyzes two chemical forms effects in metabolic markers levels, the insulin pathway IRS/Akt/P-Akt/GLUT4, and PPAR-γ, and C/EBP-β in a murine model.

**Abstract:**

Background: Human beings consume different chemical forms of iodine in their diet. These are transported by different mechanisms in the cell. The forms of iodine can be part of thyroid hormones, bind to lipids, be an antioxidant, or be an oxidant, depending on their chemical form. The excessive consumption of iodine has been associated with pancreatic damage and diabetes mellitus type 2, but the association between disease and the chemical form consumed in the diet is unknown. This research analyzes the effect of excessive iodine consumption as Lugol (molecular iodine/potassium iodide solution) and iodate on parameters of pancreatic function, thyroid and lipid profiles, antioxidant and oxidant status, the expression of IR/Akt/P-Akt/GLUT4, and transcription factors PPAR-γ and CEBP-β. Methods: Three groups of Wistar rats were treated with 300 μg/L of iodine in drinking water: (1) control, (2) KIO_3_, and (3) Lugol. Results: Lugol and KIO_3_ consumption increased total iodine levels. Only KIO_3_ increased TSH levels. Both induced high serum glucose levels and increased oxidative stress and pancreatic alpha-amylase activity. Insulin levels and antioxidant status decreased significantly. PPAR-γ and C/EBP-β mRNA expression increased. Conclusion: The pancreatic damage, hypertriglyceridemia, and oxidative stress were independent of the chemical form of iodine consumed. These effects depended on PPAR-γ, C/EBP-β, GLUT-4, and IR.

## 1. Introduction

Excessive iodine consumption has been associated with alterations in the lipid profile and hyperglycemia in the population [1,2,3,4,5,6,7,8]. Therefore, high levels of iodine have been considered a possible risk factor for the development of type 2 diabetes mellitus, characterized by the dysfunction and decrease of pancreatic beta cells [3]. In murine models, the consumption of 3.5 mg of potassium iodide (KI)/100 g body weight, which is 500 times the physiological daily dosage of iodide, for 60 days causes a hyperglycemic and hypercholesteremic status and severe destruction of the pancreatic structure [9]. In mice, the excessive consumption of 1200 or 2400 μg/L of iodide alters insulin, blood glucose levels, and the homeostasis model assessment for insulin resistance (HOMA-IR) index after 6 months [10]. In contrast, low iodine and 25(OH)D concentrations have been associated with complications in diabetic patients [8], and excessive consumption of iodine is associated with pancreatic damage in epidemiological studies [3,4,5,6,7,11]. The pancreas expresses the Na+/I− cotransporter (NIS) to capture iodides [12] and facilitates diffusion to capture molecular iodine (I_2_) [13]. High levels of I_2_ can induce oxidative damage and cellular apoptosis [13,14,15,16]. These suggest that the pancreas could be a target of excessive iodine consumption and could play a role in alterations in pancreatic function.

In Mexico, by official regulations and public health policies, the chemical form used to enrich table salt is iodate (KIO_3_). This iodate is the most thermodynamically stable of all chemical iodine forms and is without oxidative action at basic and neutral pH. Lugol (I_2_/KI solution) is an aqueous iodine solution, and I_2_ can act as a donor or an acceptor. These formulations are used to reduce iodine deficiency. Both are public health strategies for the prevention of pathologies associated with iodine deficiency. However, the lack of monitoring programs or regulation of its consumption has led to reported cases of excess iodine consumption in the population, and this could be a risk factor for the development of diabetes, hypothyroidism, and hypertriglyceridemia [3,4,5,6,7,8,9,10,11]. Studies done in models in vitro and in vivo have shown that I_2_ has antioxidant properties (in low doses) and acts as a free radical scavenger (in high doses); it also acts as an oxidant and can have cytotoxic effects by inducing apoptosis at high concentrations [15,16,17]. I_2_ can oxidize long-chain lipids and form iodolipids that are ligands of the peroxisome proliferators type γ (PPAR-γ) and regulate genes associated with apoptosis, as has been described in breast cancer cells [18]. In mature 3T3-adipocytes, high doses of Lugol cause lipolytic and cell death effects, depending on the PPAR-γ and C/EBP-α expression [19].

This study analyzes the effects of excessive consumption of two chemical forms of iodine, which are KIO_3_ or Lugol at 300 µg/mL, in drinking water for 2 months on the thyroid profile, markers of pancreatic function (i.e., insulin, glucose levels, and activity of the alpha-amylase enzyme), antioxidant and oxidative stress levels, lipid profile, and the pathway activated by glucose IRS/Akt/P-Akt/GLUT4, and the mRNA expression of transcription factors as PPAR-γ and C/EBP-β.

## 2. Materials and Methods

### 2.1. Murine Model of Iodine Excess Consumption

The murine model of Zhang et al. [17] was used. For this, 24 two-month-old male Wistar rats from the vivarium of the Faculty of Bioanalysis of the Universidad Veracruzana were used, and the research committee certified that the study subjects were healthy and fit for the experiment. The rats were kept in the vivarium of the same institution, with a 12/12 h, light–dark cycle, with food and water ad libitum. Subsequently, the animals were distributed into the following groups: (1) a control group (water as a vehicle; *n* = 8); (2) a group with excess iodate (KIO_3_; 300 μg/L in drinking water; *n *= 8); and (3) a group with excess Lugol (I_2_/KI solution; 300 μg/L in drinking water; *n *= 8). The treatments were given for 8 weeks. At the end of this time, the study subjects were euthanized with the use of sodium pentobarbital after fasting for 12 h, and blood samples were obtained by cardiac puncture for determinations of glucose, insulin, lipids, iodine, thyroid hormones, thyroid-stimulating hormone (TSH), peripheral alpha-amylase and antioxidant-oxidant activity. The experimental subjects were treated in accordance with the provisions of Mexican NOM-062-ZOO-1999, “technical specifications for the production, care, and use of laboratory animals”.

### 2.2. Determination of Iodine and Thyroid Profile in Serum

The concentration of total iodine in serum was determined using the Sandell-Kolthoff colorimetric technique. A Thermo Scientific Genesys TM 10S UV-Visible Spectrophotometer (Madison, WI, USA) was used. TSH levels were determined with the WELLS ALPCO 96 ELISA Kit (Keewaydin Drive, Salem, NH, USA). For thyroid hormones, the Free T3 AccuBind ELISA Kit, Carlsbad, CA, USA, and the Free T4 AccuBind ELISA Kit, Carlsbad, CA, USA, were used following the manufacturer’s instructions.

### 2.3. Determination of Glucose, Insulin, Alpha-Amylase and Triacylglycerols

Glucose, insulin, triacylglycerols, and cholesterol levels were determined in serum. For glucose, the Rat Glucose Assay^®^ kit (Cat 81693; Crystal Chem, Elk Grove Village, IL, USA) was used; for insulin, the Rat Insulin Kit^®,^ (Cat CC-90060; Crystal Chem, Elk Grove Village, IL, USA) was used, and for triacylglycerols kit from BioSystems^®^ (Medellin, Colombia) was used. The samples were read using the Bio Rad iMark microplate reader, Hercules, CA, USA. To determine the levels of the pancreatic amylase enzyme, the colorimetric method from the manufacturer Wiener Lab^®^, Neudorf, Austria was used.

### 2.4. Determination of Antioxidant Status

Total antioxidant status (TAS) was analyzed colorimetrically. Briefly, 1.0 mL of FRP solution (25 mL of 300 mmol/L sodium acetate pH 3.6, 2.5 mL of 50 mmol/L potassium ferrocyanide K3Fe(CN)6 and 2.5 mL of 20 mmol/L FeCl_3_ 6H_2_O) was added to the serum sample, or H_2_O, or standard with ascorbic acid and incubated for 3 min at 37 °C. The absorbance against blank H_2_O was read at 593 nm on a microplate reader (Spectramax Plus; Molecular Devices, Sunnyvale, CA, USA). The TAS in the unknown samples was determined by interpolation using a seven-point calibration curve of known amounts of ascorbic acid and expressed in mmol of ascorbic acid equivalents/liter. The enzymatic activity of superoxide dismutase (SOD) was determined by adding 2 mM of pyrogallol to a microplate for superoxide generation, 24 mM of the tetrazolium dye (MTT), 3-(4-5dimethyl thiazol 2-xl)2,5 diphenyl tetrazolium bromide, and 100 μL of the sample or blank (10 mM TRIS). The reaction was incubated for 10 min and dissolved with 100 μL of DMSO. Absorbance was measured at 570 nm in a microplate reader (Spectramax Plus; Molecular Devices, Sunnyvale, CA, USA). The activity was expressed as U/g of hemoglobin. The determination of catalase activity was carried out using 50 μL of the pancreas homogenate, and 450 μL of phosphate buffer pH 5.6 and 2.5 mL of 11 mM H_2_O_2_ were added. Kinetics was performed for 1 min at a wavelength of 240 nm in a Genesys TM 10S UV-Visible Spectrophotometer. The reaction began when hydrogen peroxide was added, so it was added, and then the absorbance reading began immediately. The reaction blank was prepared for each sample and consisted of 50 μL of homogenate, 450 μL of phosphate buffer, and 2.5 mL of injectable water. Finally, the concentration of total glutathione (GSH) was determined using Ellman’s reagent (5,5-dithio-bis-(2-nitrobenzoic acid). GSH stock solution was prepared for the calibration curve (1.5 mM, 1.25 mM, 1 mM, 0.75 mM, 0.50 mM, and 0.25 mM). The wavelength for reading was 420 nm.

### 2.5. Determination of Oxidative Stress

The determination of the oxidative stress was carried out using the TBARS assay, measured in 90 μL of lysate extract and in 70 μL of 150 mM TRIS buffer with a pH of 7.5. A total of 300 μL of the mixture was combined with 0.4% thiobarbituric acid. Then, 20% acetic acid with a pH of 3.0 was added to 90 μL of each sample. All samples were heated at 100 °C for 45 min in a thermal transfer. The samples were cooled on ice, and 1% potassium chloride KCl was added. After centrifugation, 180 μL of supernatant was added and measured at 532 nm in a microplate reader (Spectramax Plus; Molecular Devices, Sunnyvale, CA, USA). The results are expressed in absorbance units per 0.1 mL of nmol/mg protein.

### 2.6. Immunoblotting

The pancreas tissue was lysed with 0.1 mL of RIPA Buffer (1% Triton X-100, 150 mM NaCl, 0.1% SDS, 50 mM Tris-HCl pH 7.6, 10 mM EDTA, 1 mM PMSF, and 10 µg/mL of Cocktail Protease Inhibitor). Protein concentration was quantified with the Bradford method. Lysates with 50 µg of protein under reducing conditions were subjected to SDS-PAGE (10% gels) followed by Western blotting with antibodies against β-actin (Cell Signaling Biotechnologies, Danvers, MA, USA), IR (ab5500), GLUT4 transporter (ab654) from Abcam, Cambridge, MA, USA, total Akt (mAb#4691), PPAR-γ (D69), C/EBP-β (G-10, sc-166258), β-actin (13 × 10^5^), and phosphorylated Akt in ser-473 (antibody #9271) from Cell Signaling Biotechnologies. The antibodies were used at 1:1000 in the TBST buffer with 10% BSA. β-actin was from Cell Signaling Biotechnologies 13 × 10^5^. The membrane was incubated at 4 °C with primary antibodies. Then, membranes were washed in TBST buffer and incubated with the secondary antibody (cell signaling 65-6120) diluted in TBST and BSA at 10% (1:4000) for 1 h. The signal was revealed using 1:1 with Amersham ECL Prime Western Blotting (Thermo Fisher Scientific Inc., Waltham, MA, USA) with Gel-Doc (BioRad, Hercules, CA, USA). The ImageJ Version 1.52a program (Wayne Rasband, Bethesda, MD, USA) was used to calculate the relative OD ratio. The insulin receptor, Glut-4, and total Akt were normalized to β-actin levels and p-Akt with the total Akt protein.

### 2.7. Determination of mRNA by RT-PCR

TriReagent was used in the total RNA extraction (Sigma-Aldrich, Saint Louis, MO, USA). Cells were spiked with TriReagent and incubated at room temperature. Then, the cells were transferred to 1.5 mL Eppendorf tubes. Chloroform was added, and the samples were incubated for 10 min. Then centrifugation was conducted at 12,000 rpm at 4 °C. The aqueous phase was recuperated, and isopropanol was added. After centrifugation at 12,000 rpm, the pellet was washed three times with 70% cold ethanol and evaporated. The pellet was resuspended in sterile water, and the quantification of total RNA was conducted in a spectrophotometer. Electrophoresis in 2% agarose gel staining with ethidium bromide was conducted to corroborate the RNA integrity. The ReverTra Ace^®^ qPCR RT master mix was used to synthesize mRNA to cDNA, and Quantitative real-time PCR was conducted with a SensiFAST™ SYBER^®^ No-ROX Kit (Sigma-Aldrich, Saint Louis, MO, USA) in Piko Real-Time real PCR equipment (Thermo Sci, Waltham, MA, USA). The primers to PPAR-γ forward were 5′-TGCCAGTTTCGATCCGTAGA-3′ and reverse 5′-AGGAGCTGTCATTAGGGACATC-3′. The primers to C/EBP- forward were 5′GCGCGAGCGCAACAACA 3′ and reverse 5′TGCTTGAACAAGTTCCGCAG3′. The primers to GAPDH forward were 5′-GGCCATCCACAGTCTTCTGG-3′ and reverse 5′-ACCACAGTCCATGCCATCACTGCCA-3′. Denaturation temperature was carried out at 94 °C for 3 min. The next 40 cycles were 35 s at 94 °C, 30 s at 58–60 °C, and 30 s at 72 °C. The GAPDH housekeeping gene was used to normalize the data as an internal control, and the relative expression fold change was calculated for each target transcript with the following formula: 2-(ΔCt), where ΔCt was Ct (target genes) − Ct (housekeeping gene). 

### 2.8. Statistical Analysis

The data were presented in frequencies, means, and standard error. The sample distribution was analyzed with the Kolmogorov–Smirnov test, so two-way ANOVA and Holm–Sidak’s post-hoc test were used because the data distribution was normal. A confidence level of 95% was used, and *p* < 0.05 was considered statistically significant compared to the control group.

## 3. Results

### 3.1. Total Iodine Levels and Thyroid Profile in Rats Exposed to 300 μg/L of Lugol and KIO_3_

Treatments with 300 μg/L of KIO_3_ and Lugol in the drinking water increased the concentration of total iodine in serum, as reported in the literature. These results were statistically significant with respect to the control group (*p* < 0.05). However, no significant changes were observed in the levels of free thyroid hormones (T3 and T4) compared to the control group. However, TSH levels increased in the groups exposed to iodine excess compared to the control group (*p* = 0.008; Table 1).

### 3.2. Effect of Iodine Excess Consumption in Drinking Water on the Metabolism of Glucose and Pancreatic Function

In Table 2, the results show that insulin levels decreased significantly in both groups with excess iodine compared to the control group; this decrease was greater in the group administered with KIO_3_. Likewise, serum glucose levels increased in both groups with iodine consumption excess compared with the control group, with a significant level of *p* < 0.05. The study also found an increase in the levels of the enzymatic activity of pancreatic alpha-amylase in the groups treated with iodine in comparison with the control group (*p* < 0.05). These data indicate pancreatic damage in the groups with excess iodine consumption, with both chemical forms.

### 3.3. Effect of Iodine Excces Consumption on Lipid Profile

In Table 3, the results show a significant increase in the levels of triacylglycerols in both groups exposed to Lugol and KIO_3_ compared to the control group (*p* < 0.05). There were no significant changes in total cholesterol levels compared to the untreated control group.

### 3.4. Iodine Excces Consumption Alters the Antioxidant and Oxidant Status

Table 4 shows the effect of excess iodine in drinking water in the pancreatic tissue. Related to the total antioxidant status, the results show a decrease in the two groups administered with iodine excess (Lugol and KIO_3_) compared to the untreated control group (*p* < 0.0001). These effects were accompanied by low activity of superoxide dismutase and catalase enzymes, while the total glutathione content decreased only in the group exposed to KIO_3_, compared to the control group. In relation to oxidative stress, the results show an increase in malondialdehyde levels, which was significant only in the group treated with 300 μg/L (*p* = 0.0005 vs. control group).

### 3.5. Excessive Iodine Consumption Affects the Expression of GLUT4, PPAR-γ, and C/EBP-β in Pancreatic Tissue

The results show that iodine excess consumption as Lugol or KIO_3_ in drinking water during 8 weeks increased PPAR-γ and decreased IR significantly in comparison with the control group. The Akt, Akt/P-Akt ratio did not change in comparison with the control group. However, GLUT4 expression increased significantly in both the KIO_3_ and Lugol groups compared with the control group (Figure 1).

### 3.6. Excessive Iodine Consumption Affects the Expression of PPAR-γ and C/EBP-β in Pancreatic Tissue

The results show that PPAR-γ and C/EBP-β mRNA expression was increased in the groups with iodine excess consumption as Lugol or KIO_3_ in drinking water after 8 weeks in comparison with cthe ontrol group (*p* < 0.05; Figure 2).

## 4. Discussion

In this study, treatments with excess iodine, either via KIO_3_ or Lugol at 300 µg/mL, administered in the drinking water increased total blood iodine levels and TSH levels. These results are in agreement with the model reported by Yoshida et al., 2014 [20]. Related to the thyroid profile, the results show that excess iodine in drinking water with both chemical forms did not alter the levels of free T3 and T4, although TSH levels did increase, suggesting subclinical hypothyroidism. In this regard, it has been reported that murine models that consume up to 183 mg/kg/day of KIO_3_ do not alter TSH levels [21,22]. However, the use of 300 mg/kg/day in this study did alter this variable. The increase of TSH levels due to excess iodine stimulates thyrotropin-releasing hormone (TRH), and this, in turn, stimulates the release of TSH. In this study, the increase of TSH was observed in subjects exposed to excess iodine; it did not increase the levels of free thyroid hormones, which coincides with some studies that relation this effect to the thyroid compensatory mechanism, which is characterized by enlargement of the thyroid gland (acute phase of the Wolff–Chaikoff effect) and that, after the withdrawal of excess iodine, TSH levels can decrease or hypothyroidism may occur [23].

Related to the redox state, iodine excess in both chemical forms decreases the antioxidant status. In this study, total antioxidant status, catalase, and superoxide dismutase, as well as total glutathione, were decreased compared with the control untreated group. This is consistent with existing research, where it was shown that the antioxidant status decreases and is postulated to be associated with pancreatic beta cell death [24] or with pancreatic fibrosis [25]. We also observed an increase in levels of oxidative stress only with KIO_3_. In this regard, it has been shown that excess iodine intake leads to a high concentration of glutathione and malondialdehyde in organs such as kidneys and is consistent when iodine is administered in excess long-term in Wistar rats [26]. In pancreatic beta cells and adipocytes, high doses of Lugol induce oxidative stress, although the most sensitive were the pancreatic beta cells to Lugol [16].

This experimental study showed that excessive consumption of iodine, regardless of the chemical form, causes hypertriglyceridemia, pancreatic damage, and KIO_3_ oxidative stress. Regarding the latter, an increase in glucose levels, a decrease in insulin, and an increase in the activity of the alpha-amylase enzyme was observed, which is associated with pancreatic damage. In epidemiological studies has been associated excessive iodine consumption as a risk factor for type 2 diabetes mellitus and hyperglycemia [1,2,3,4,6,8,9,11]. In Mexico, salts and flours are mandatorily fortified with iodate, and various reports indicate the overconsumption of these foods, although epidemiological studies are necessary to determine the effects of excessive iodine consumption in the population. In addition to this, it is known that excessive iodine consumption is associated with the development of hypothyroidism, alterations in the levels of triacylglycerides, and glucose, the latter being signs associated with chronic non-communicable diseases, main causes of death in the adult population [6]. In vitro studies have shown that adipocytes and pancreatic beta cells exposed to low levels of Lugol promote glucose transport, and this is dependent on PPAR-gamma expression, p-Akt, and GLUT4; however, high levels of iodine cause cell death by caspase-3 activation in pancreatic beta cells [16]. In epidemiological studies, the excessive consumption of iodine is related to the development of alterations in glucose metabolism, and the authors postulate that this effect is due to the deregulation of thyroid hormones caused by the increase in serum TSH levels [3,4,7,17].

The study by Sarkar et al. is consistent with what was reported in this study, showing that excess iodine (0.5 mg of KI/100 g of body weight) in male Wistar rats for 60 days alters glucose levels. In the groups administered with iodate, glucose values of 150 mg/dL were reported, significantly higher than the controls, which showed an average of 110 mg/dL [21]. These results are similar to those of this study. Furthermore, the deficiency or excessive consumption of iodine has been associated with the risk of presenting insulin resistance [9]. Similarly, in our study, excess iodate caused high insulin levels and a decrease in IR, which correlates with other studies that show that low levels insulin levels and low levels of IR are associated with pancreatic fibrosis, insulin resistance, and alterations in insulin gene regulation [3,4,7,8,9,10].

Although it has been reported that supplementation with Lugol (0.2 mg/kg/day) can contribute to the reduction of pancreatic damage caused by inducing agents [27], our study shows that consumption of excess Lugol alters the profiles associated with pancreatic function, so it is important to regulate iodine consumption to avoid unwanted effects on other iodine-capturing organs such as the pancreas. Excess iodine, in its two chemical forms, increases the activity of the pancreatic alpha-amylase enzyme. In this regard, the increase in the activity of this enzyme is associated with the presence of pancreatitis at values equal to or greater than 1500 IU/Dl [28,29,30]. One of the factors in the development of pancreatitis is a hyperglycemic state, as shown in our model of excess iodine. Thus, pancreatitis is accompanied by oxidative stress and decreased antioxidant status. Iodine excess has the capacity to generate reactive oxygen species, which interact with membrane phospholipids, causing lipoperoxidation, and this phenomenon with excess of iodine in pancreatic beta cells is associated with the increase in pancreatic amylase and oxidative stress [16]. 

Previous reports have highlighted the importance of iodine in lipid metabolism because, in adipocytes exposed to Lugol, the content of lipids and total triglycerides decreases, and the release of glycerol increases. This is credited to the induction of lipolysis mediated by activation of caspase-3 through the regulation of PPAR-α and C/EBP-α [19], coupled with the fact that high levels of triacylglycerides are considered a risk factor associated with cardiovascular damage.

The protein increase of GLUT4 suggests that pancreatic damage caused by excess iodine consumption could regulate the expression of this transporter and then contribute to high glucose levels. In this regard, previous data showed that glucose transport depends on GLUT4 expression, and this could be related to PPAR-γ activation because this transcription factor directly regulates the expression of GLUT4 protein [3,18], and iodine as Lugol induces PPAR-γ expression on mature adipocytes 3T3-L1 and beta-pancreatic cells [16,19]. Previously, it has been reported that PPAR-γ regulates GLUT4 through promoter region −66/+163 pb that contains response elements to PPAR (PPRE) [31]. The role of PPARγ in pancreatic βeta cells is not fully understood, and in contrast, PPAR-γ agonists protect the pancreatic βeta cells from cell death and restore their functionality [32], while PPAR-γ agonists can also decrease insulin secretion in diabetic animal models [33]. Therefore, it is necessary to carry out studies to elucidate the role of the transcription factors of the PPAR family and its association with the regulation of glucose metabolism mediated by iodine chemical forms.

It has also been reported that hypothyroidism can increase the expression of GLUT4, which correlates with TSH increases, in this study, and elevated levels of TSH are associated with subclinical hypothyroidism, which could be contributing to altering glucose levels and pancreatic function [34]. It has also been reported that GLUT4 located in the pancreas is regulated by high glucose levels in pancreatic β-cells [35]. In trophoblast cells, iodine deficiency decreases the expression of C/EBP-β and is accompanied by an increase in ROS and nitric oxide associated with poor migration and trophoblast differentiation [36].

The biological effects of iodine on the pancreas have been previously documented. From these previous reports, it should be noted that it depends on multiple conditions. For example, in the presence of molecules with toxic activity for the pancreas, such as methimazole, treatment with molecular iodine at doses of 2.0 mg/kg (not in excess) acts as a protective agent and captures free radicals. Therefore, physiological or normal concentrations of iodine contribute to the function and protection of the pancreas [27]. This is supported by in vitro studies in mature adipocytes and pancreatic beta cells exposed to low and high levels of Lugol, which induces an influence on the transport of glucose in mature adipocytes and insulin secretion in pancreatic beta cells, as well as cell death dependent on oxidative stress and caspase-3 activation in these cells [16,19]. 

## 5. Conclusions

The results suggest that excessive iodine consumption as Lugol or KIO_3_ causes either oxidative stress, hypertriglyceridemia, damage, and loss of function of the pancreas, with alterations in glucose metabolism and insulin resistance.

At the molecular level, this study observed an increase in the GLUT4 protein and a decrease in IR. These effects are associated with high expression of the PPAR-γ transcription factor involved in the iodine action mechanism and a decrease in C/EBP-β.

It is necessary to determine whether the transport mechanisms for each chemical form, as well as its chemical properties, are associated with its harmful effects.

Epidemiological studies are needed to understand the effects of excessive consumption of iodine on health, and in vitro and in vivo experimental studies are needed to elucidate its mechanism of action in the pancreas. 

## Figures and Tables

**Figure 1 biology-13-00466-f001:**
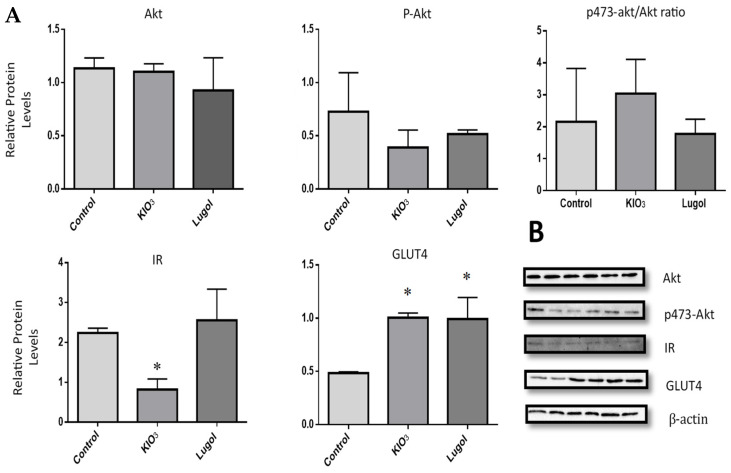
Effect of Lugol and KIO_3_ excess in drinking water given after 8 weeks on the regulation of IR/AKT/P-Akt/GLUT-4 in Wistar rats. (**A**) densitometric analysis. (**B**) Protein bands from a representative Western blot. (1) Control group (water as a vehicle; *n *= 8); (2) group KIO_3_ (300 μg/L in drinking water; *n *= 8); and (3) group Lugol (300 μg/L in drinking water; *n *= 8). The treatments were given for 8 weeks. At the end of this time, the study subjects were euthanized, and pancreas tissue was lysed with RIPA buffer. The protein extracts were subject to SDS-PAGE 10%, then the blots were incubated with primary and secondary antibodies, as described in the methodology section. Detection was performed with the chemiluminescence kit. Data were normalized with β-actin levels and its reported relative protein levels to the IR/total Akt/GLUT-4 and p473-Akt with total Akt protein (ratio levels). Data represented mean ± SD, *n *= 3. * *p* < 0.05.

**Figure 2 biology-13-00466-f002:**
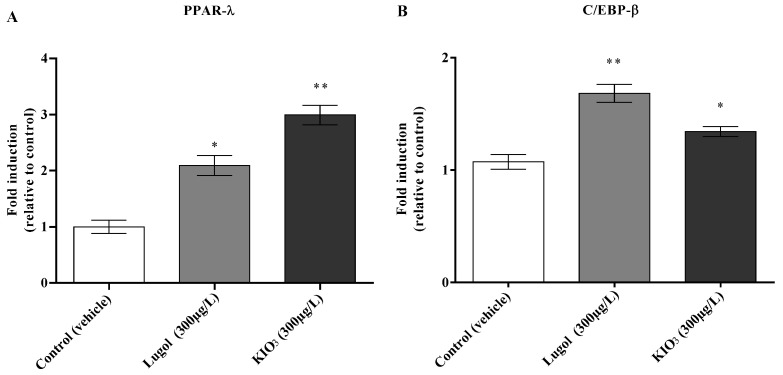
Effect of Lugol and KIO_3_ exccess consumption in drinking water after of eigth weeks on the regulation of PPAR-γ and C/EBP-β mRNA expression. (**A**) PPAR-γ fold induction. (**B**) C/EBP-β fold induction. (1) Control group (water as a vehicle; *n *= 8); (2) group with Lugol (300 μg/L in drinking water; *n *= 8). (3) group KIO_3_ (300 μg/L in drinking water; *n *= 8); the treatments were given for 8 weeks. At the end of this time, the study subjects were euthanized, and pancreas tissue was lysed with RIPA buffer. Total RNA was extracted with TRIreagent, as described in the methodology section. The housekeeping gene GAPDH was an internal control and used to normalize the change of expression and calculate the relative fold change of expression, and the transcripts of target genes were quantified as 2-(ΔCt), where ΔCt was Ct (target genes) − Ct (housekeeping gene). Data represented mean ± SE, *n *= 8. * *p* < 0.05; ** *p* < 0.01.

**Table 1 biology-13-00466-t001:** Effect of consumption of KIO_3_ and Lugol (300 μg/L) administered in drinking water on the thyroid profile of the study groups.

Group	Serum Iodine Levels (µg/mL)	TSH (mUI/L)	Free T3 (nmol/dL)	T4 Free (nmol/dL)
Control (*n *= 8)	0.5 ± 0.03	0.45 ± 0.01	0.17 ± 0.0008	0.6 ± 0.018
Lugol (300 μg/L) (*n *= 8)	12.9 ± 0.04 ^a^	0.50 ± 0.02 ^b^	0.18 ± 0.0007	0.65 ± 0.014
KIO_3_ (300 μg/L) (*n *= 8)	14.3 ± 0.09 ^a^	0.59 ± 0.02 ^b^	0.18 ± 0.0008	0.62 ± 0.015

The mean ± standard error (SE) is presented. The comparison of groups was performed with two-way ANOVA. ^a^ vs. the control, *p* < 0.05; ^b^ *p* = 0.008.

**Table 2 biology-13-00466-t002:** Effect of iodine excess consumption (300 µg/L) of Lugol or KIO_3_ on glucose level and pancreatic function.

Parameter	Control (*n *= 8)	Lugol (300 μg/L) (*n *= 8)	KIO_3_ (300 μg/L) (*n *= 8)
Serum insulin (mg/dL)	85 ± 3.2	62.8 ± 2.2 ^a^	47.6 ± 1.1 ^a^
Serum glucose (mg/dL)	83.5 ± 3.8	169.9 ± 1.3 ^a^	148.1 ± 1.7 ^a^
Pancreatic alpha-amylase (UI/L)	992.6 ± 15.8	1215.4 ± 13.5 ^a^	1115.3 ± 20.9 ^a^

The mean ± standard error (SE) is presented. The comparison of groups was performed with two-way ANOVA, and *p* < 0.05 was considered statistically significant. ^a^ vs. control, *p* < 0.05.

**Table 3 biology-13-00466-t003:** Effect of excess iodine on lipid profile.

Parameter	Control (*n *= 8)	Lugol (300 μg/L) (*n *= 8)	KIO_3_ (300 μg/L) (*n *= 8)
Triacylglycerols (mg/dL)	132.6 ± 3.2	150.0 ± 1.7 ^a^	155.0 ± 2.1 ^a^
Total cholesterol (mg/dL)	78.8 ± 4.2	69.8 ± 1.6	70.4 ± 0.5

The mean ± standard error (SE) is presented. The comparison of groups was performed with two-way ANOVA, and *p* < 0.05 was considered statistically significant. ^a^ vs. control group, *p* < 0.05.

**Table 4 biology-13-00466-t004:** Effect of excess iodine on antioxidant-oxidant status.

Parameter	Control (*n *= 8)	Lugol (300 μg/L) (*n *= 8)	KIO_3_ (300 μg/L) (*n *= 8)
TAS (mm of ascorbic acid equivalents/L)	193.2 ± 47.8	100.4 ± 4.9 ^a^	108.3 ± 0.8 ^a^
SOD activity (U/g of hemoglobin)	1050 ± 232	224.1 ± 56.3 ^a^	195 ± 17 ^a^
Catalase activity (H_2_O_2_/mg de protein)	2095 ± 49.5	1056.2 ± 9.9 ^a^	1035.9 ± 6.6 ^a^
Total Glutathione (mm cysteine/mg protein)	0.04 ± 0.005	0.07 ± 0.01	0.002 ± 0.0004 ^a^
Oxidant Stress (µM MDA/mg protein)	1.4 ± 0.3	1.5 ± 0.1	29.5 ± 6.6 ^a^

The mean ± standard error (SE) is presented. The comparison of groups was performed with two-way ANOVA, and *p* < 0.05 was considered statistically significant. ^a^ vs control group, *p* < 0.05.

## Data Availability

All data are contained within the article or Appendix A.

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
