# Peer review of "Excess Iodine Consumption Induces Oxidative Stress and Pancreatic Damage Independently of Chemical Form in Male Wistar Rats: Participation of PPAR-γ and C/EBP-β"

_biology, 2024, doi:10.3390/biology13070466_

Round 1

Reviewer 1 Report

Comments and Suggestions for Authors

a

Rezultat prijevoda

Author Response

response to the reviewer 1:

Thank you very much for taking the time to review this manuscript. Please find the detailed responses below and the corresponding revisions/corrections highlighted/in track changes in the re-submitted file.

 Reviewer: Abstract: the results should go in the same order as stated in the methods.

Answer: results and methods are in the same order.

Reviewer: keywords in alphabetical order

Answer: was corrected.

Reviewer: Statistical analysis: "The data were presented in frequencies, means, and standard error"; it means that there was normal distribution for all variables, if that is correct please state it that way.

Answer: in methodology section in statistical analysis was added the follow sentence:

The sample distribution was analyzed with the Kolmogorov-Smirnov test, so one-way ANOVA and Holm-Sidak's post-hoc test were used because the data distribution was normal.

Reviewer: footnotes below tables and figures - all abbreviations should be mentioned

Answer:  the abbreviations were mentioned in tables and figures

Reviewer: line 268, qg/mL

Answer: the line 268 was corrected

Reviewer: mark the tables in the text with Arabic numerals

Answer: the table marks were corrected and Arabic numeral were added.

Thanks

Reviewer 2 Report

Comments and Suggestions for Authors

dr Arbez-Evangelista Cristian and coworkers, in this manuscript, analyzed the effect of excess iodine consumption on different parameters (thyroid and lipid profile, antioxidant and oxidant status, protein levels of insulin receptor (IRS)/Akt/P-Akt(ser-473) /GLUT4, and transcription factors PPAR-γ, and CEBP-β).

The paper is well written. The experimental design and methods are simple and straightforward.

Just one reccomendation for the authors: In the introduction it is necessary to include the chemical characteristics of the two substances used and in the conclusions, since slight differences are observed in the response of the mice to the two experimental substances, it is necessary to emphasize the reasons for such differences and not write generically "independently of its chemical form".

Finally, in figure 1, perhaps the authors should verify the statistics of the "relative protein levels of IR", observing bar graph and standard deviation, it seems strange to me that KIO3 is not statistically significant.

Author Response

Response to the reviewer 2

Thank you very much for taking the time to review this manuscript. Please find the detailed responses below and the corresponding revisions/corrections highlighted/in track changes in the re-submitted file.

Reviewer: Just one recommendation for the authors: In the introduction it is necessary to include the chemical characteristics of the two substances used and in the conclusions, since slight differences are observed in the response of the mice to the two experimental substances, it is necessary to emphasize the reasons for such differences and not write generically "independently of its chemical form".

Answer: the chemical characteristics from both iodine forms (in red) where added in the introduction and conclusions sections

 with this sentences:

Introduction

The pancreas expressing the Na+/I− cotransporter (NIS) to capture iodides [12], and facilitated diffusion to capture molecular iodine (I2) [13],

In Mexico, by official regulations and public health policies the chemical forms used to enrich table salt is iodate (KIO3), this iodate is the most thermodynamically stable of all chemical iodine forms, without oxidative action at basic and neutral pH; and lugol (I2/KI solution) is an aqueous iodine solution and I2 can act as a donor or an acceptor, these formulations are used to revert iodine deficiency, both are public health strategies to the prevention of pathologies associated with iodine deficiency.

In the conclusion section was added this sentence:

 It is necessary to determine whether the transport mechanisms for each chemical form, as well as its chemical properties, are associated with its harmful effects.

Reviewer: Finally, in figure 1, perhaps the authors should verify the statistics of the "relative protein levels of IR", observing bar graph and standard deviation, it seems strange to me that KIO3 is not statistically significant.

Answer: The figure 1 was corrected, the statistic was verified and effectively, the protein levels of the insulin receptor was significantly between KIO3 and control group. The asterisk was added in the graphic and the legend.

The result was added in graphical abstract, in results with this sentencedecreasing IR significantly in comparison with control group”, in dicussion: “Similarly, in our study, excess iodate caused high insulin levels, and decrease of IR, which correlates with other studies that show that low levels Insulin levels and low levels of IR are associated with pancreatic fibrosis, insulin resistance and alterations in insulin gene regulation”. Conclusion: The results suggest that excessive iodine consumption as Lugol or KIO3 causes either oxidative stress, hypertriglyceridemia, damage and loss function of the pancreas, with alterations in glucose metabolism, and insulin resistance.

At the molecular level was observed an increase of GLUT4 protein, and decrease IR. These effects are associated with high expression of the PPAR-γ transcription factor in-volved in the iodine action mechanism, and decrease of C/EBP-β.

Thanks

Reviewer 3 Report

Comments and Suggestions for Authors

In this paper, the author dived into the mechanism behind high iodine consumption and type 2 diabetes and hyperglycemia. The authors first hypothesized the relevance of pancreatic cells in the excess iodine consumption-induced health risks. With this hypothesis, the author collected comprehensive evidence based on an established murine model of excessive iodine consumption. The evidence includes assessing glucose metabolism and pancreatic function, lipid profile, and antioxidant activities. Additionally, the author further dived into molecular mechanisms, and showed the PPAR-γ, and C/EBP-β expressions are upregulated transcriptionally upon excessive iodine exposure.

The research is well designed with sound logic. The reference is appropriate. A few comments can be found: 

1.    Table 1 and table 2 : for the reported values, aligning decimal places and/or significant figures. For example, in table 1, the TSH in the control group was reported to 2 d.p., while the Lugol group was reported to 1 d.p.

2.    Table 1 to 4: providing justification for statistical analysis methodology. Specifically, why one-way ANOVA is chosen.

3.    Discussion section: including the discussion for transcriptional impact of excessive iodine intake. It is very interesting that PPAR-gamma is upregulated upon iodine exposure. Given PPAR-gamma is a key transcription factor with a broad spectrum of substrates, it would be interesting to explore the additional transcriptional impact of excessive iodine intake, to expand our understanding.

Author Response

Response to reviewer 3

Thank you very much for taking the time to review this manuscript. Please find the detailed responses below and the corresponding revisions/corrections highlighted/in track changes in the re-submitted file.

Reviewer:    Table 1 and table 2 : for the reported values, aligning decimal places and/or significant figures. For example, in table 1, the TSH in the control group was reported to 2 d.p., while the Lugol group was reported to 1 d.p.

Answer:  Was corrected the values from the Tables 1 and 2.

Reviewer: 2.    Table 1 to 4: providing justification for statistical analysis methodology. Specifically, why one-way ANOVA is chosen.

Answer: By mistake, a one-way ANOVA was written, when a 2-way ANOVA statistical analysis was done and carried out to compare the means of more than 2 variables between the 3 experimental groups. This was corrected in each table and in section methodology.

Reviewer: 3.    Discussion section: including the discussion for transcriptional impact of excessive iodine intake. It is very interesting that PPAR-gamma is upregulated upon iodine exposure. Given PPAR-gamma is a key transcription factor with a broad spectrum of substrates, it would be interesting to explore the additional transcriptional impact of excessive iodine intake, to expand our understanding.

Answer: was added this sentence in discussion section:

Previously has been reported that PPAR-γ regulate GLUT4, through promoter region −66/+163 pb that contains response elements to PPAR (PPRE) [31]. The role of PPARγ in pancreatic βeta cells is not fully understood, and contradictory, a mean, PPAR-γ agonists protect the pancreatic βeta cells from cell death and restore its functionality [32]. In contrast, PPAR-γ agonists can also decrease insulin secretion in diabetic animal models [33], therefore, it is necessary to carry out studies to elucidate the role of the transcription factors of PPAR family, and its association in the regulation of glucose metabolism mediated by iodine chemical forms.

Thanks